# Fluorescence lifetime Hong-Ou-Mandel sensing

Ashley Lyons [1] ✉, Vytautas Zickus [1,2], Raúl Álvarez-Mendoza [1], Danilo Triggiani [3], Vincenzo Tamma [3,4], Niclas Westerberg [1], Manlio Tassieri [5] & Daniele Faccio [1] ✉

Fluorescence Lifetime Imaging Microscopy in the time domain is typically performed by recording the arrival time of photons either by using electronic time tagging or a gated detector. As such the temporal resolution is limited by the performance of the electronics to 100's of picoseconds. Here, we demonstrate a fluorescence lifetime measurement technique based on photon-bunching statistics with a resolution that is only dependent on the duration of the reference photon or laser pulse, which can readily reach the 1–0.1 picosecond timescale. A range of fluorescent dyes having lifetimes spanning from 1.6 to 7 picoseconds have been here measured with only ~1 s measurement duration. We corroborate the effectiveness of the technique by measuring the Newtonian viscosity of glycerol/water mixtures by means of a molecular rotor having over an order of magnitude variability in lifetime, thus introducing a new method for contact-free nanorheology. Accessing fluorescence lifetime information at such high temporal resolution opens a doorway for a wide range of fluorescent markers to be adopted for studying yet unexplored fast biological processes, as well as fundamental interactions such as lifetime shortening in resonant plasmonic devices.

Fluorescence lifetime imaging microscopy (FLIM) measures the exponential decay time of fluorophores excited by an ultrafast source. FLIM is used across the bio-imaging community to provide key information about local biological environments as it can be dependent on pH, temperature, viscosity, and chemical concentrations. The lifetimes of commonly used fluorophores are within the range of 100's of picoseconds to nanoseconds. However, lifetimes can shorten down to 10's of femtoseconds depending on the nature of the fluorophore[1]. As such, access to higher resolutions enables the use of femtosecond scale fluorophores[2,3], coupling to plasmonic devices which can increase quantum yield[4,5], and exploration of vibrational and rotational modes that act on the femtosecond scale[6].

In the time domain, the lifetime is measured by a combination of a single-photon detector and an electronic timing mechanism such as either gating the detector or by assigning each individual photon an arrival time through time-correlated single photon counting (TCSPC). The resolution is then typically limited in the 50–100 ps scale or longer[7]. This is due, for example, to the instrument response function (IRF) of the detector, which is limited by effects such as electron diffusion[8].

Nonlinear optical gating methods where the fluorescent signal is combined with a laser pulse in a nonlinear frequency-conversion crystal, have demonstrated femtosecond scale resolution for fluorescence lifetime measurements[9–13]. However, the overall low efficiency of the nonlinear effects requires measurements to be performed with high power lasers and to have a limited wavelength flexibility for a given system, due to phase matching considerations and choice of nonlinear crystals. For instance, Kerr-gating is an alternative nonlinear

[1]School of Physics and Astronomy, University of Glasgow, Glasgow G12 8QQ, UK. [2]Department of Laser Technologies, Center for Physical Sciences and Technology, LT-10257 Vilnius, Lithuania. [3]School of Mathematics and Physics, University of Portsmouth, Portsmouth PO1 3QL, UK. [4]Institute of Cosmology and Gravitation, University of Portsmouth, Portsmouth PO1 3FX, UK. [5]James Watt School of Engineering, University of Glasgow, Glasgow G12 8QQ, UK. ✉e-mail: ashley.lyons@glasgow.ac.uk; daniele.faccio@glasgow.ac.uk

optical method that can achieve femtosecond-scale resolutions, but with efficiencies reaching up to 50% at best[14–16]. Alternatively, streak cameras can achieve sub-picosecond resolutions albeit at the cost of significant hardware cost and limited dynamic range, specifically when operated at high temporal resolutions[17]. Similarly, single-photon superconducting nanowire detectors (SNSPDs) have recently been reported with few-picosecond temporal resolutions[18,19].

Hong-Ou-Mandel (HOM) is a two-photon interference effect whereby two indistinguishable photons that meet at a beamsplitter will leave via the same output port[20]. This photon bunching leads to a characteristic "HOM dip" in the photon coincidence counts measured by the second-order correlation function $g^{(2)}(\tau)$, as a function of the relative delay time, $\tau$. Originally proposed for measuring sub-picosecond time intervals, the use of HOM interferometry for photon time-of-flight sensing has since been adopted for measuring optical delays due to birefringence[21], and path-delay measurements[22–24] with attoseconds time-scale resolution[25,26]. Recently, HOM interferometry has been applied to molecular spectroscopy, demonstrating an ability to measure dephasing times on the 100 fs time-scale from an organic dye in solution[27]. Notably, two-photon HOM interference can be observed between completely independent light sources as first demonstrated by Rarity et al.[28] and can even be extended to incoherent light sources. This is well illustrated by the work performed by Deng et al.[29] showing HOM interference between photons generated by a quantum dot and photons collected from the Sun.

Fluorescence lifetime can also be measured by monitoring the photon anti-bunching when pumped by two femtosecond pulses with a relative temporal delay[30], this approach is, however, restricted to isolated single molecules. Photon bunching has also been used for higher densities of molecules[3,31] with schemes that produce an auto-correlation of the fluorescence signal. Recovering the actual lifetime from the auto-correlation requires the solution of an ill-posed inverse problem[32]. In this work, we propose HOM interference between fluorescence photons and secondary reference photons and demonstrate experimentally the application in the classical case where the reference photon is replaced by an attenuated laser pulse. Our approach, which we call fluorescence lifetime Hong-Ou-Mandel (FL-HOM), offers the benefit of measuring directly the fluorescence lifetime signal by using a reference photon or laser pulse that is much shorter than the fluorescence lifetime. FL-HOM is able to operate at low photon numbers and does not rely on any optical nonlinear components. We demonstrate the potential of this approach by

characterising a range of fluorescence lifetimes in the picosecond range, including those from so-called molecular rotors that are sensitive to the environment viscosity. Previous measurements with these rotors were limited to high viscous systems (i.e., long lifetimes). With FL-HOM, their lifetime can be resolved in the ps regime, thus enabling 'contactless nanorheology' measurements of very low viscosity fluids.

## Results
### Experimental layout
The setup is made of a Mach-Zehnder interferometer using two outputs from a single femtosecond laser source running at an 80 MHz repetition rate (Coherent Chameleon Discovery NX). The first output, which we label "Excitation Pulse" (EP), has a fixed wavelength at 1040 nm and 140 fs pulse duration. This is frequency-doubled in a BBO crystal to 520 nm and focused on to the fluorescent sample using a Nikon 60×, 0.7 NA microscope air objective. The fluorescence is collected along the same beam path in a confocal geometry, and spatially filtered by coupling into a single mode polarisation maintaining fibre. An interference filter is placed before the fibre so as to select a spectral band around 660 nm with two different bandwidth options (details below). The second laser output, which we label "Reference Pulse" (RP), synchronised to the EP, and is tuned to the fluorescent 660 nm wavelength and coupled into a single mode fibre with the same spectral filtering as the fluorescence arm. The fibre coupler is mounted on a translation stage so as to control the relative path delay between the two interferometer arms. Both interferometer arms are then directed to a fibre-coupled 50:50 beamsplitter and photon correlations are measured between the two beamsplitter outputs by a pair of Single Photon Avalanche Diode (SPAD) detectors and a commercial time-correlated single photon counting (TCSPC) unit (ID Quantique ID801).

### Lifetime sensing
The intensity profile of the fluorescent emission is given by the following convolution integral:

$$I_{FP}(t) = I_{ex}(t) * F_{fl}(t), \tag{1}$$

where $I_{ex}(t)$ is temporal intensity distribution of the excitation pulse, $F_{fl}(t) = A e^{-t/\mu}$ is the fluorescent single exponential decay response of the sample, with amplitude $A$ and decay constant (i.e. fluorescence lifetime) $\mu$. Here, we focus on the limit where the excitation pulse is of short duration with respect to $\mu$ such that $I_{FP}(t) \simeq F_{fl}(t)$. The general

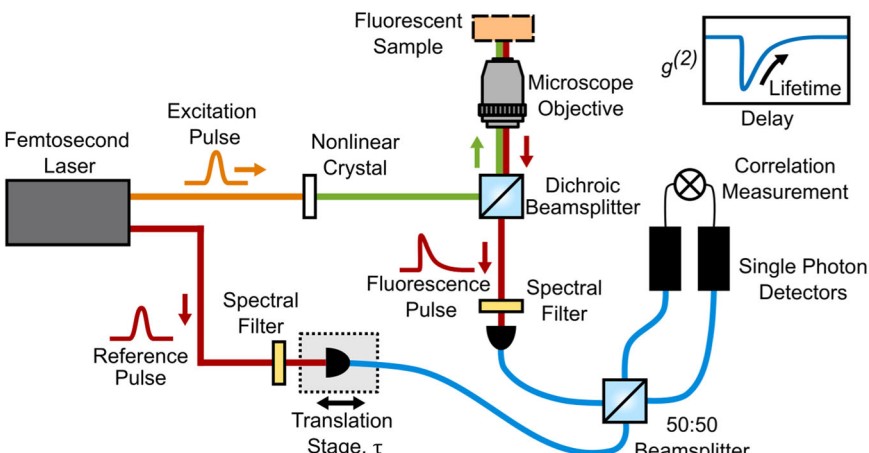

**Fig. 1 | Schematic layout of the experimental setup consisting of a Mach-Zehnder interferometer within one arm of which we place the fluorescent sample.** Not shown in the figure are spectral filters to remove the 1040 nm pump before the microscope objective and various lenses for re-sizing the beam. Red lines indicate the reference and emitted fluorescence beam-paths at 660 nm, orange shows the 1040 nm pump beam for the SHG, and green shows the resulting 520 nm excitation beam. An illustration of the measured second-order correlation function, $g^{(2)}(\tau)$, is shown in the top right inset from which the lifetime is estimated.

form of the normalised second-order intensity correlation function is:

$$g^{(2)}(t') = \frac{\langle I_1(t)I_2(t+t')\rangle}{\langle I_1(t)\rangle\langle I_2(t+t')\rangle}, \tag{2}$$

where $I_1(t)$ is the intensity measured at one detector at time $t$ and $I_2(t)$ is the intensity measured at the second detector at a time $t+t'$. From Eq. (2), it is possible to calculate the normalised number of coincident photon pair events (which we define as those occurring within some time window of the same detection time i.e. $t' = 0$) as a function of the optical delay, $\tau$, between the two arms of the interferometer controlled by a translation stage (see Fig. 1):

$$C(\tau) \sim 1 - 4C_0(\sigma)e^{-\tau/\mu}, \tag{3}$$

where $\sigma$ is the temporal duration of the reference photon or laser pulse, and $C_0$ is a constant that encapsulates the visibility of the interference depending on the amplitude of the two pulses, their durations, and the reflection and transmission constants of the beamsplitter. Here we have used that the reference pulse is much shorter than the lifetime (i.e. $\sigma \ll \mu$), and that $\tau > 0$. A full derivation of $C(\tau)$ can be found in the SI.

From Eq. (3) it is possible to observe that the lifetime can be measured directly from the number of coincident events. We also note that the resolution can be controlled by changing the pulse duration of the reference and/or excitation pulses, which can be achieved by using spectral bandpass filters before the detectors.

## Measurements of picosecond fluorescence decay

Figure 2a shows an example of a normalised FL-HOM trace for a sample consisting of 4-DASPI dye[33]. To ensure a high level of indistinguishability between the two interferometer arms, a narrowband 0.6 nm

FWHM spectral filter (Alluxa 660-0.5-OD6) was used in each arm. This has the overall effect of increasing the interference visibility at the expense of reducing the achievable temporal resolution. Indeed, a wider bandwidth filter of 10 nm showed, under the same conditions, a maximum visibility on the order of ten times smaller dependent on the lifetime (see below). Typical photon count rates were of the order $10^6$ photons/s with a photon coincidence count rate of ~$14 \times 10^3$ photons/s outside of the HOM dip. The data shown in Fig. 2a was obtained with a long acquisition time of 2 s per delay point, for a total measurement time of approximately 40 minutes, and the number of coincident events was normalised by the total number of photon counts to account for fluctuations in laser power and fibre coupling. The step size of the translation stage equates to an optical delay of 16.7 fs. Figure 2a also shows a fit to our model without any approximations about the length of the reference pulse (i.e. before Eq. (3)) and the measured IRF, from which we recover a lifetime of $7.22 \pm 0.04$ ps. Previous work in ref. 33 measured a lifetime of 11 ps for 4-DASPI in water. While this is relatively close to our measurement, we attribute our shorter lifetime to potential differences in the temperature, pH, and emission wavelength chosen, all of which can affect the lifetime. To help validate our approach, we also conducted a measurement of 4-DASPI in ethanol which is known to produce a longer lifetime. From this, we observed a lifetime of $65 \pm 1.2$ ps which is in close agreement with previous measurements of 62 ps[34] (see Supplementary Material). In order to investigate and ensure the capability of our approach, we significantly over-sampled the data and purposely measured for long times, thus resulting in a much lower noise level than a typical time domain FLIM measurement. As a means of comparison, Fig. 2b shows a similar measurement, but using a total acquisition time of only 3.5 s. By using a Markov Chain Monte Carlo approach for the lifetime retrieval, our model still returns a reliable lifetime of 6.8 ps. A full distribution of lifetimes derived under these conditions (i.e., 3.5 s total acquisition

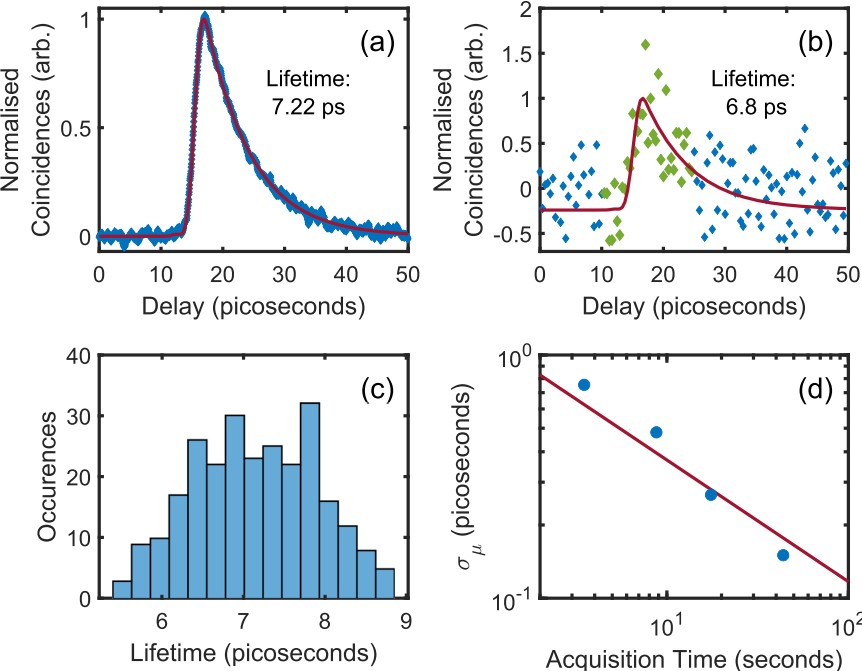

**Fig. 2 | Lifetime retrieval from the numerical model for 4-DASPI. a** Fluorescence lifetime curve for the 4-DASPI sample (the curve has been inverted and normalised so as to appear more familiar compared to standard FLIM curves). Blue shows experimental data (2 s acquisition per delay point) and red depicts a fit from our model described in the Supplementary Material using Equation S11. **b** The same measurements with a 3.5 s total acquisition time, retrieving the same approximate lifetime. based on fitting to the points shown in green. **c** Shows the statistical distribution of retrieved lifetimes for N = 260 measurements (3.5 s acquisition time each), indicating a mean of 6.8 ps and a standard deviation of 0.7 ps. **d** The standard deviation of the lifetime estimation as a function of the total measurement acquisition time. Blue points are the experimental data points and the red line illustrates the expected reduction in uncertainty proportional to the square root of the measurement time.

time, 260 individual measurements) is shown in Fig. 2c, which has a standard deviation of $\sigma_\mu = 0.7$ ps. In Fig. 2d we report $\sigma_\mu$ versus total measurement time. This data were obtained by taking multiple short exposures of 50 ms for each optical delay point and summing them to obtain longer acquisition times. This was then repeated over multiple iterations to statistically evaluate the uncertainty of the measured lifetime (typical measurements and fits for each acquisition time can be found in the SI). In Fig. 2d the red line indicates a gradient of −0.5 i.e. a line proportional to the square root of the total measurement time which we expect to observe given that we have a set of independent and statistically equivalent measurements. It is also useful to determine how the number of coincident events themselves scale as a function of the measurement time. We evaluate this (see SM) finding that the signal-to-noise ratio improves proportionally to $N/\sqrt{1+2N}$. For $N \gg 1$ the SNR $\sim \sqrt{N}/\sqrt{2}$, indicating that coincidence counting brings a penalty of $\sqrt{2}$ in the SNR compared to direct intensity measurements.

To further demonstrate the FL-HOM approach we repeated the measurements for two alternative fluorescent dyes i.e., Allura Red and Pinacyanol Iodide. Figure 3 shows the related data, where we plot the logarithm of the normalised coincident events. From the analysis of these measurements, we retrieve lifetimes of 6.23 (4-DASPI), 1.60 (Allura Red) and 3.66 (Pinacyanol Iodide) picoseconds. As the lifetimes measured here are close to the effective IRF (see below), we validated our shortest lifetime measurement using the Allura Red against our MCMC retrieval. From this, we found a lifetime of 1.51 ps indicating that simple tail fits provide a reasonable lifetime estimation at these short timescales. We also observed a variation in the lifetime by changing the temperature using the 4-DASPI sample, showing a lifetime range of 5-8.25 picoseconds (see SI).

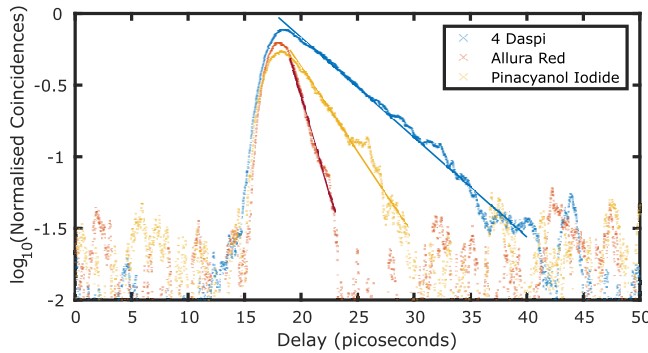

**Fig. 3 | Lifetime estimation of different samples.** The natural log of the (inverted) normalised coincidence events is shown for comparison with typical FLIM data. Crosses indicate measured data points and solid lines show a linear fit to the exponential tail.

## Interference visibility

The maximum achievable interference visibility is dependent on the overlap integral between the fluorescence and reference pulses. Figure 4a shows the normalised number of coincident events for the Allura Red sample with both a 0.6 nm and a 10 nm bandpass filter, keeping all other conditions fixed. We measure the IRFs with these two filters to be 2.08 ps and 0.17 ps FWHM, respectively (see 'Methods'). In the case of the broadband filter, the temporal overlap between the two pulse always remains low due to the relatively large difference between the 140 fs reference pulse and the much longer 1.6 ps duration of the lifetime. With the narrowband filter on the other hand, the maximum temporal overlap becomes much larger as a 0.6 nm bandwidth corresponds to a transform-limited pulse of order 1 ps duration. We also show (dashed black line) the normalised single photon counts for one of the detectors in our setup under conditions where we achieved maximum HOM interference visibility. No interference fringes can be observed confirming that we are operating in a regime where there is no first-order interference. In Fig. 4b, we further model how the HOM interference visibility varies with the reference photon or laser pulse duration by evaluating Eq. (3). We observe that there is a trade-off between the temporal resolution and the FL-HOM visibility with optimal visibility being achieved when the pulse duration is 0.7× the lifetime. For longer/shorter pulse durations the pulses become less distinguishable in the temporal domain thereby reducing the visibility.

## Molecular rotor lifetime as a probe for viscosity

As an example of FL-HOM capability, we show that lifetime measurements of rotor molecules with picosecond resolution enables contact-free nanorheology over a broad range of viscosities. Rheology is the study of flow of matter, and its applications have revealed valuable information on biological specimens[35] at different time and length scales, down to intracellular dynamics at microscales[36]. Recent work has also revealed the need for contact-free nanoscale rheology to better understand the impact of viscosity on carbon capture and microbe populations in the oceans[37]. Nanorheology has been enabled in cells using atomic force microscopy, where a metal tip is placed in 'contact' with the sample[38]. However, such an approach is not viable for intracellular or remote measurements.

Rotor molecules interact with the surrounding medium through their rotational degrees of freedom. The formation of twisted intramolecular charge transfer (TICT) states upon photoexcitation leads to competing de-excitation pathways: fluorescence emission and non-radiative de-excitation from the TICT state. Since TICT formation is viscosity-dependent, the emission intensity and lifetime of molecular rotors depend on the solvent's viscosity[39,40]. However, intensity-based measurements are influenced by (i) the optical properties of the surrounding environment, (ii) the dye concentration and (iii) solvent-dye interactions, thus requiring a calibration of

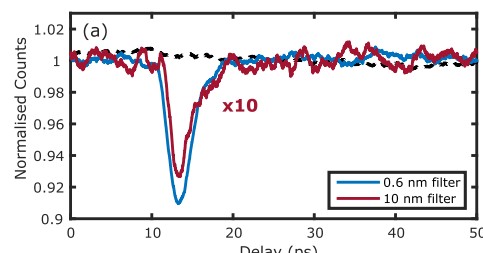

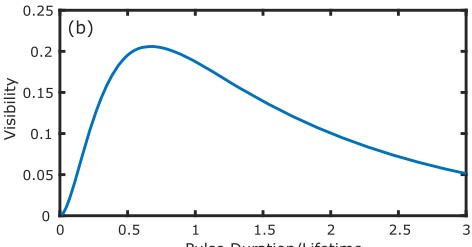

**Fig. 4 | Effects of the optical bandwidth on the interference visibility and resolution. a** The effect of spectral filtering for two different bandwidths is shown for Allura Red fluorescence. The wider bandwidth results in a higher temporal resolution measurement at the expense of reducing the interferometric visibility.

The signal from the 10 nm broadband filter has been scaled by a factor of 10x for comparison. The black dashed line shows the normalised single photon counts at one of the detectors for the 0.6 nm filter measurement. **b** Model of the visibility as a function of the ratio between the pulse duration and fluorescence lifetime.

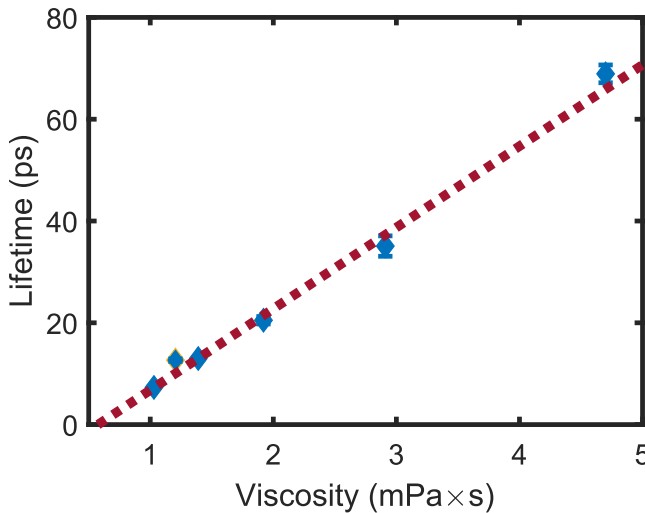

**Fig. 5 | FL-HOM lifetime measurement of 4-DASPI as a viscosity probe.** Glycerol is added to the solution increase the viscosity, starting from that of pure water. More than an order of magnitude dynamic range is observed in the lifetime with no appreciable change in the photon count rate, indicating a high level of sensitivity to the viscosity. Uncertainties in the lifetime are shown although for most point, the error bars are smaller than the data point and thus not visible.

## Discussion

In Fig. 2b we demonstrate that our FL-HOM system can retrieve a lifetime measurement as low as 6.8 ps within 3.5 s measurement duration; with potentials to be significantly improved. The SPADs were kept at a maximum count rate of $10^6$ photons per second to avoid pileup effects with a probability of detecting a photon of only around 1% for each of the 80 MHz rate laser pulses and a 0.016% probability of detecting a correlated pair event. Bearing in mind that the probability of measuring a coincidence scales with the product of the photon numbers at each detector, decreasing the repetition rate of the laser while keeping the average photon number constant can lead to an increase in the number of measured coincidences. For example, if a repetition rate of 0.1 MHz is used with an average (photon starved) 10% probability to measure a photon at each detector from each pulse, the measurement time can be reduced by a factor of ~10x implying that the measurements in Fig. 2b could be obtained in sub-second timescales.

The high temporal resolution of FL-HOM provides the possibility to study faster fluorescent dynamics such as those from dyes with femtosecond-scale lifetimes[2,3]. This allows for a wider range of markers to be used for FLIM, thereby also increasing the dynamic range. Individual molecular processes can also be investigated with transitions relating to the vibrational modes commonly falling in the femtosecond regime[6]. Although longer lifetime fluorophores generally have a greater quantum yield than shorter lifetime probes, interaction with plasmonic modes have also shown a strong increase in the fluorescence intensity at the expense of shortening the lifetime[4,5]. We also note that as our method directly measures the temporal distribution of the fluorescent light, it allows also for the measurement of bi-exponential and more generalised fluorescent decays. It is also not restricted to measurements of single molecules and in this way, is complementary to techniques such as those shown in ref. 30.

It is worth mentioning that our approach is related, but somewhat different, to 'fluorescence lifetime correlation spectroscopy' (FLCS)[49]. In FLCS, rapid fluctuations of the fluorescence intensity are recorded in time in a Hanbury-Brown Twiss interferometer, i.e. a single beamsplitter is placed *after* the sample and correlations between the two outputs are recorded without the presence of any additional reference photons. These correlations can be resolved within the duration of the fluorescence decay but are still limited in resolution by the IRF of the SPAD sensors, similarly to standard FLIM approaches.

Finally, our experiments are implemented with a classical reference laser pulse. One could use a single photon generated for example by a parametric downconversion (PDC) or a quantum dot source. It has previously been shown that the visibility of the HOM interference between a single photon source and, for example, an attenuated laser[28] can approach unity where an equivalent measurement with classical light would be limited to a $g^{(2)}$ of 0.5. Furthermore, a PDC source provides greater flexibility in terms of tuning the reference wavelength to the fluorescence emission by operating at non-degenerate photon pair wavelengths controlled through the phase-matching of the generation crystal.

## Methods

### Fluorescent dye sample preparation

4-DASPI (Sigma-Aldrich 336408) and Allura Red AC (Sigma-Aldrich 458848) were dissolved in purified water at concentrations of ≈ 2.7 3 mM and ≈5.04 mM, respectively, and the Pinacyanol iodide (Alfa Aesar H31540) was dissolved in MeOH (Sigma-Aldrich 1.06002) at the concentration of ≈0.12 mM. The samples were measured at room temperature, in 10 mm path length UV fused quartz cuvettes (Thorlabs CV10Q35E). We note that in all cases, the probability of re-absoprtion in these samples is low due to the relatively large Stokes shift. We also confirm that there is a low probability of interaction between

the measurement system to a specific solvent. Lifetime measurements instead, provide the opportunity of a calibration-free approach[41]. However, previous studies have been limited to the measurement of very high viscous fluids (i.e., nearly pure glycerol, which is circa thousand times more viscous than water) as only in this regime the rotor molecules (such as 4-DASPI) have sufficiently long lifetimes for these to be measured and/or resolved with ~100 ps of standard FLIM approaches. For instance, 4-DASPI has been used for viscosity measurements by using both intensity[42] and sub-nanosecond lifetime measurements[43] for which the shortest lifetime measurements were of order of 700 ps.

We used 4-DASPI dye in water and glycerol/water mixtures to explore a range of low viscosity values (i.e., of the order of mPa·s), while maintaining a constant pH and temperature (see 'Methods'). The viscosity within cellular environments can lie between 1–400 mPa·s or even higher[44–46]. Here we focus on the lower 1–5% of this range to demonstrate the high sensitivity of the technique. The viscosity of each mixture was measured by means of conventional bulk-rheology measurements performed with a commercial rheometer (see 'Methods'). Figure 5 shows FL-HOM lifetime measurements as a function of viscosity. We observe more than an order of magnitude change in lifetime, with no appreciable change in the photon count levels thus revealing that FL-HOM lifetime measurement is a much more sensitive metric than intensity measurements at such low viscosity values. Interestingly, at lower viscosity values, it is only the 2.8 ps reference pulse duration after the filter (our effective IRF) that is limiting the viscosity measurement to a value as low as 0.0065 mPa·s, which is more than two orders of magnitude lower than that of water (which is $\eta$ ~ 1mPa·s). Moreover, FL-HOM returns a resolution of <1% and is thus competitive to more established microrheology measurement techniques such as multiple particle tracking or optical tweezers[47,48]. At relatively high viscosity values, the noise in the decay function measurement at long times is the limiting factor affecting the measurements, yet the method still returns a ~2−4% error in viscosity measurements. Notably, for longer lifetimes than those explored here, one would simply switch to using the standard time-correlated-single-photon techniques by analysing the data at just one detector, with no changes required in the setup.

fluorophores while in the excited state. A discussion of this can be found in the SM.

## Glycerol/water mixtures

Samples for viscosity measurements were prepared by varying the weight concentration of Glycerol (Sigma-Aldrich G5516) in purified water and 4-DASPI (Sigma-Aldrich 336408) dye. 4-DASPI was kept at constant concentration of 0.1 wt% in all samples. The glycerol/water-4-DASPI solutions were then titrated to a pH of ≈7.0 while continuously mixed.

## Effective IRF measurement

To estimate the IRF, we perform a linear auto-correlation of the 660 nm reference pulse. A 50:50 beamsplitter is placed before the single mode fibre seen in Fig. 1a and the reflection is coupled into another fibre. This then replaces the input from the fluorescence arm into the same fibre-coupled beamsplitter to ensure spatial mode matching. The coincident photon counts at one of the detectors is measured as a function of the delay stage position and a low pass filter is applied to remove any residual first-order coherence fringes and retrieve the envelope function. From this, we measure a FWHM of the 2.08 ps and 0.17 ps for the 0.6 nm and 10 nm bandpass filters, respectively.

## Markov Chain Monte Carlo lifetime estimation for short acquisition times

The lmfit package in Python was first used to fit the experimental data to our model by solving the nonlinear Least Squares regression problem with the Levenberg-Marquardt algorithm[50]. The parameter space around the best-fit solution was explored by means of the Affine Invariant Markov Chain Monte Carlo (MCMC) Ensemble sampler that is implemented in lmfit via the Minimizer.emcee() method, which allowed us to refine the initial Least Squares estimate. The sampling rate of the lifetime curves that were deconvolved using this approach was one-tenth of that of the curves obtained using long acquisition times, and simultaneously a smaller delay region was sampled where the experimental noise is lowest to reduce total acquisition time. For the lifetime curve shown in Fig. 2, this implies that the peak was sampled with only 35 delay points, resulting in acquisition time of 3.5 s.

## Conventional rheology

Bulk rheology measurements were performed on each explored sample to determine the ground-truth viscosity. These were performed by using a stress controlled rheometer (Anton Paar Physica MCR 302 Instruments) equipped with a cone-plate measuring system (CP60-1-SN42255). The temperature was controlled by means of a Peltier system connected to a water bath. The fluids' viscosity was measured by performing a flow curve test at shear rates varying from 50 s$^{-1}$ to 500 s$^{-1}$. For each sample the viscosity had a constant value within the explored range of shear rates.

## Data availability

All data used to produce the plots shown in this article is available at https://doi.org/10.5525/gla.researchdata.1524.

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

## Acknowledgements

The authors acknowledge financial support from the Royal Academy of Engineering under the Chairs in Emerging Technology and Research Fellowships schemes, and the U.K. Engineering and Physical Sciences Research Council (Grants No. EP/T002123/1, EP/T00097X/1). D.F., A.L. and M.T. acknowledge financial support from the U.K. Engineering and Physical Sciences Research Council (Grants No. EP/X035905/1). N.W. wishes to acknowledge support from the Royal Commission for the Exhibition of 1851. V.Z. received funding from European Social Fund (project No 09.3.3-LMT-K-712-23-0132) under a grant agreement with the Research Council of Lithuania. V.T. acknowledges support from the Air Force Office of Scientific Research under award number FA8655-23-1-7046.

## Author contributions

The concept for the work was developed by A.L., V.Z. and D.F. Experimental work was undertaken by A.L., V.Z., R.A.M. while D.T., VT. and N.W. developed the theoretical framework. M.T. provided valuable insights for the rheology sections of the work as well as ground truth viscosity measurements. The project was supervised by A.L. and D.F.

## Competing interests

The authors declare no competing interests.
