## [Peer Review File · Nature Communications]

Fluorescence Lifetime Hong-Ou-Mandel SensingREVIEWER COMMENTS

Reviewer #1 (Remarks to the Author):

The manuscript by Lyons et al. reports about a new technique to measure fluorescence lifetimes based on Hong-Ou-Mandel sensing. The authors state that the typical measurement of fast fluorescence lifetimes below 100 ps is not feasible by time correlated single photon counting, due to the Instrument Response function (IRF) (the authors state this incorrect with "Impulse Response Function", which suggests that solely the excitation pulse is responsible for the limited time resolution). However, in bulk or ensemble measurements the fluorescence lifetime can also be measured with a Streak-Camera down to a few picoseconds. Maybe the authors can also compare their approach to this technique? Is their approach then still beneficial in some way?

Later in the introduction the authors also compare their work to ref. 19, in which isolated single molecules have been measured by antibunching (not bunching as the authors state), if I understand this work correctly. The authors might want to double-check this. In this context, I believe that the work of the authors is complementary, because it is limited to ensembles of molecules and can not be used at the single-molecule level, due to the limited signal, which is mainly given by narrow band-pass filters, which must be used to observe interference effects. On the other hand, the work of ref. 19 is solely limited to single molecules.

Regarding this point I would be interested how their measurement is dependent on the concentration of the solution? How low or how high a concentration can interference effects still be measured?

Besides these points I have to say that I like this approach a lot and it seems to me like an innovative idea. However, the authors might also point out the necessary count rate to perform this measurement and as mentioned above it seems to me that this approach is not applicable to single molecules, due to the limited photon budget available. But maybe the authors can prove me wrong on this, which would strengthen the applicability.

In summary I like this idea and I hope that the authors have very good arguments to overcome this problem or I have to suggest publication in a more specialized journal for spectroscopy.

Reviewer #2 (Remarks to the Author):

The authors propose and demonstrate a novel fluorescence lifetime measurement technique based on photon-bunching statistics with Hon-Ou-Mandel (HOM) interferometer. Sub-picosecond time resolution is achieved and an exciting application of contactless nanorheology in low-viscosity fluids is provided. Overall, I find the work very exciting and highly relevant, and it should be considered for publication in Nature Communications but only after the authors carefully address the following concerns and questions.

1. The Introduction is rather weak, and it does not make a good case for the need of picosecond fluorescence lifetime measurement. The authors should expand their Introduction far beyond just saying “However, lifetimes can shorten down to 10’s of femtosecond depending on the nature of the fluorophore [1].”
2. Comparison between their FL-HOM and existing sub-picosecond fluorescence lifetime measurement techniques is insufficient.
 - a. For its comparison with nonlinear optical gating, the authors claim that “*FL-HOM does not rely on any optical nonlinear components and is well suited for low photon numbers.*” While the reviewer agrees that their FL-HOM does not require a high-power gate (or reference) pulse, it is not clear whether FL-HOM is indeed more suited in photon starved conditions. As the authors mentioned in the Introduction, up to 50% nonlinear conversion efficiency can be achieved given a high-power gate pulse. The main issue of nonlinear optical gating is the low photon efficiency due to the large mismatch between the gate pulse duration and the fluorescence lifetime. From this perspective, it seems like FL-HOM has a similar issue as the authors showed in Figure 4: there is a trade-off between time resolution and interferometric visibility (and thus measurement sensitivity). The authors should provide more justifications including some photon budget calculation to support their argument.
 - b. Similarly, it is not acceptable that the authors only have a sentence that compares their FL-HOM with FL-HBT [19-21]. First of all, it is the reviewer’s understanding that their claim “*However, this approach provides the auto-correlation of the fluorescence signal, which requires the solution of an ill-posed inverse problem to recover the actual lifetime.*” only applies to [20, 21] but not to [19]. In fact, it is stated in [19] that fluorescence lifetime can be measured without the need for any deconvolution. Naively, the reviewer would think the authors’ FL-HOM has an edge over FL-HBT [19] because their FL-HOM can be applied to fluorophores with complicated energy-level diagrams. If this is true, the next question is whether FL-HOM can be applied to study double exponential fluorescence lifetime. The reviewer does not expect new measurements, but some discussions will come a long way.
3. The authors show three FL measurements of different fluorophores in Figure 3 and temperature-dependent FL of 4-DASPI in Figure S1. How do their FL-HOM measurement results compare to those using existing methods? Without references, it is hard to evaluate the accuracy of their new technique.
4. According to the derivation, Eq (3) is valid only for the case $\sigma \ll \mu$. On the other hand, the optimum visibility happens when $\sigma = 0.7 * \mu$ as shown in Figure 4b. Now, one can argue that σ should not be too much smaller than μ for a reasonable measurement signal-to-noise ratio. For this reason, the reviewer thinks it is important to at least include the first-order perturbation to their analytic model to understand the effect of finite σ/μ .
5. Another question about Figure 4b. It is the reviewer’s understanding that the maximum visibility does not reach 0.5 because of the pulse shape mismatch between the reference

(Gaussian) and the fluorescence signal (exponential). If this is true, have the authors considered other reference pulse shape to get closer to 0.5 visibility?

6. For Figure 5, there are only three data points in the low viscosity regime (1-2 mPa x s) where the FL-HOM is most powerful at. The authors should take a few more data points to strengthen their error analysis for this important low viscosity regime.
7. At the end of the paper, the authors claim that a PDC source can better match the fluorescence emission wavelength and provide a factor of 2 increase in the HOM visibility. Didn't the authors already match the wavelength by using narrowband filters? It is not clear why PDC source will help and where the factor of 2 comes from.

Reviewer #3 (Remarks to the Author):

The authors report on a novel method to measure fluorescence decays with exquisite picosecond time resolution. It is based on the Hong-Ou-Mandel effect which causes two photons incident onto a beamsplitter at the same time from different directions to be both reflected and transmitted in the same direction, i.e. together, as illustrated here

Michał Jachura and Radosław Chrapkiewicz, "Shot-by-shot imaging of Hong–Ou–Mandel interference with an intensified sCMOS camera," *Opt. Lett.* 40, 1540-1543 (2015)

<https://doi.org/10.1364/OL.40.001540>

In the present manuscript, one photon comes from the fluorescence decay, and one photon comes from the excitation pulse. One of them is registered in one detector, whereas the other detector is guaranteed not to detect a photon. It may be worth a mention that only one of the two Hong-Ou-Mandel photons is detected, to make this clearer.

Anyway, this causes a dip in the second order correlation function g_2 , if I understand this correctly. The delay between excitation pulse photon and fluorescence photon is changed, and the depth of the dip in g_2 is plotted versus the delay, and this represents the fluorescence decay. As the optical width of the excitation pulse is femtoseconds, a picosecond time resolution can be achieved with this method.

This is one of the highest time resolutions reported, similar to streak cameras

ANTHONY J. CAMPILLO AND STANLEY L. SHAPIRO, Picosecond Streak Camera Fluorometry-A Review *IEEE JOURNAL OF QUANTUM ELECTRONICS*, VOL. QE-19, NO. 4, APRIL 1983 585, doi: 10.1109/JQE.1983.1071909

and a recently reported short linear super-conducting nanowire

Korzh, B., Zhao, QY., Allmaras, J.P. et al. Demonstration of sub-3 ps temporal resolution with a superconducting nanowire single-photon detector. *Nat. Photonics* 14, 250–255 (2020).

<https://doi.org/10.1038/s41566-020-0589-x>

Hadfield, R.H. Superfast photon counting. *Nat. Photonics* 14, 201–202 (2020).

<https://doi.org/10.1038/s41566-020-0614-0>

The authors put this approach to use by studying short fluorescence lifetime fluorophores, including a fluorescent molecular rotor. They show its fluorescence lifetime increase in the 10s of picosecond range with increasing viscosity, as expected for these types of fluorophores.

This is an ingenious idea reminiscent of pump probe approaches which have been proposed for fluorescence lifetime measurements

Buist, A.H., Müller, M., Gijbbers, E.J., Brakenhoff, G.J., Sosnowski, T.S., Norris, T.B. and Squier, J. (1997), Double-pulse fluorescence lifetime measurements. *Journal of Microscopy*, 186: 212-220. <https://doi.org/10.1046/j.1365-2818.1997.2090773.x>

and FLIM

MÜLLER, M., GHARALI, R., VISSCHER, K. and BRAKENHOFF, G. (1995), Double-pulse fluorescence lifetime imaging in confocal microscopy. *Journal of Microscopy*, 177: 171-179. <https://doi.org/10.1111/j.1365-2818.1995.tb03547.x>

The manuscript is well written and straight-forward to follow, with good contextualisation and discussion of alternative methods to measure fluorescence decays.

This is an excellent piece of work that will be of interest to fluorescence spectroscopists and probably microscopists, the FLIM community and the applied quantum optics research community.

I only have some minor comments:

1) Is the Nikon 60x 0.7 NA an air objective?

2) Prior DASPI and pinacyanol decay work can be found here

C.J. TREDWELL and CM. KEARY, PICOSECOND TIME RESOLVED FLUORESCENCE LIFETIMES OF THE POLYMETHINE AND RELATED DYES, Chemical Physics, Volume 43, Issue 3, 1 November 1979, Pages 307-316 [https://doi.org/10.1016/0301-0104\(79\)85199-X](https://doi.org/10.1016/0301-0104(79)85199-X)

W. Sibbett, J.R. Taylor, Passive mode locking in the blue spectral region, Optics Communications 46(1), 32-36, 1983, [https://doi.org/10.1016/0030-4018\(83\)90025-1](https://doi.org/10.1016/0030-4018(83)90025-1)

Michael Maus, Els Rousseau, Mircea Cotlet, Gerd Schweitzer, Johan Hofkens, Mark Van der Auweraer, and Frans C. De Schryver, Arnd Krueger, New picosecond laser system for easy tunability over the whole ultraviolet / visible / near infrared wavelength range based on flexible harmonic generation and optical parametric oscillation Rev Sci Instrum 72, 36–40 (2001)
<https://doi.org/10.1063/1.1326930>

Do the authors' lifetime values agree with the ones quoted in these references?

3) p3, first column, line 3 – the resolution is the “time resolution”, I presume? Maybe explicitly say “time resolution”

4) p3, second column, line 4, full time distribution, fig 2c. How many measurements is this? The integral of the distribution? A few hundred?

5) p3, second column, line 14, full time distribution, fig 2d. In TCSPC, the standard deviation depends on the square root of the number of counts (or the time) due to Poisson statistics. So doubling the acquisition time in TCSPC only improves the standard deviation by a factor of $\sqrt{2}$, whereas here it halves it. Is this right? If so, this would be worth a mention.

6) p4, second column, short DASPI lifetime. As $\phi = k_r \tau$, with ϕ the fluorescence quantum yield, k_r the radiative rate constant and τ the lifetime, short lifetime fluorophores usually have a low fluorescence quantum yield, i.e. they are dim. Long lifetime

fluorescent molecular rotors like bodipys are not only easier to measure, but they are also brighter.

7) p5, fig 5. Calibration plots for fluorescent molecular rotors are often presented on a double logarithmic scale (as log lifetime vs log viscosity) with a straight-line fit yielding a gradient around 2/3. Such a plot would also be useful here, maybe as an inset.

8) p5, column 2, line 20 “lifetime decay of fluorescence” doesn’t make any sense, the authors probably mean “fluorescence decay”, so that it should read something like “...can be resolved within the duration of the fluorescence decay....”

9) p5, column 2, Methods, fluorophore concentration. mM concentrations are very high, the solutions must have been noticeably coloured. According to

Chandrasekhar, S. 1943. Stochastic problems in physics and astronomy. *Rev. Mod. Phys.* 15:1–89. <https://doi.org/10.1103/RevModPhys.15.1>

the average distance between individual fluorophores is $0.55/[c]^{1/3}$ with $[c]$ the concentration in particles/volume. At 5 mmol/l, this yields 3.8 nm.

The average distance the fluorophore diffuses while in the excited state is given by $l = \sqrt{2 D \tau}$, where D is the translational diffusion coefficient, given by $D = kT / 6 \pi \eta r$, with k Boltzmann constant, T absolute temperature, η viscosity and r the radius of the fluorophore. For $T=293$ K, $\eta = 1$ cp, $r = 0.5$ nm, we have $D = 430 \text{ nm}^2/\mu\text{s}$, in good agreement with experimental values for Coumarin 343 ($550 \text{ nm}^2/\mu\text{s}$) and rhodamine 6 ($400 \text{ nm}^2/\mu\text{s}$) according to

Viplove Tyagi et al 2022 *Methods Appl. Fluoresc.* 10 044007, <https://iopscience.iop.org/article/10.1088/2050-6120/ac896c>

Thus $l = 0.1$ nm, and, on average, the fluorophores do not interact during the lifetime of the excited state, and re-absorption of the emitted fluorescence is probably also low due to the large Stokes shift of DASPI. Fig 5 should then really only represent viscosity sensitivity, and not interaction or self-absorption. This may perhaps be worth a brief discussion?

10) ref 5, details?

11) SI, line 6 after eq S1b. Do the authors mean eq 1 in the main text?

12) SI eq S2. What is R and T?

13) Si fig S1, State solvent used to dissolve 4-DASPI

Reply to Referees

We like to first of all thank the reviewers for their valuable comments. We have carefully considered each point individually and made alterations to our manuscript in reply to each comment. We believe that this has improved the clarity of our work and, on the whole, made the manuscript stronger as well as inciting some interesting discussion amongst the authors. Below, we list each of the points raised by the reviewers and our replies. We indicate the reviewer's comments in black, our responses in *blue*, and any alterations to the manuscript in *red*.

We hope these changes now put our manuscript in a position where the reviewers see it as suitable for publication in Nature Communications.

I. REVIEWER 1

The manuscript by Lyons et al. reports about a new technique to measure fluorescence lifetimes based on Hong-Ou-Mandel sensing. The authors state that the typical measurement of fast fluorescence lifetimes below 100 ps is not feasible by time correlated single photon counting, due to the Instrument Response function (IRF) (the authors state this incorrect with "Impulse Response Function", which suggests that solely the excitation pulse is responsible for the limited time resolution). However, in bulk or ensemble measurements the fluorescence lifetime can also be measured with a Streak-Camera down to a few picoseconds. Maybe the authors can also compare their approach to this technique? Is their approach then still beneficial in some way?

Streak cameras can indeed be used for picosecond resolution lifetime measurements however there are several significant drawbacks. Firstly, streak cameras are expensive with high resolution models costing of more than €100k, closer to 200k. whereas our approach relies only on relatively cheap linear optical components. Secondly, high temporal resolutions can only be achieved with a streak camera over a limited dynamic range. See, for example, the C13410 high dynamic range models from Hamamatsu [https://www.hamamatsu.com/content/dam/hamamatsu-photonics/sites/documents/99_SALES_LIBRARY/sys/SHSS0021E_C13410.pdf] which list a dynamic range of 10000:1 at 100 ps resolution, of 1000:1 when operated at 5 ps resolution and it then falls off very rapidly after that. The authors are aware from discussions with sales representatives that single or sub-picosecond resolutions can only be achieved at specific average photon numbers. We agree with the reviewer that streak cameras should be mentioned to give a more thorough overview of alternative methods and as such have added the following to the introduction.

"Alternatively, streak cameras can achieve sub-picosecond resolutions albeit at the cost of significant hardware cost and limited dynamic range, specifically when operated at high temporal resolutions."

In addition, we would also like to thank the reviewer with regards to their comment about the IRF acronym. We originally chose "Impulse Response Function" as a general term used within signal processing, however we appreciate that "Instrument Response Function" is more widely used in reference to TCSPC and FLIM. We have therefore changed it in the manuscript.

Later in the introduction the authors also compare their work to ref. 19, in which isolated single molecules have been measured by antibunching (not bunching as the authors state), if I understand this work correctly. The authors might want to double-check this. In this context, I believe that the work of the authors is complementary, because it is limited to ensembles of molecules and can not be used at the single-molecule level, due to the limited signal, which is mainly given by narrow band-pass filters, which must be used to observe interference effects. On the other hand, the work of ref. 19 is solely limited to single molecules.

Regarding this point I would be interested how their measurement is dependent on the concentration of the solution? How low or how high a concentration can interference effects still be measured?

We would like to thank the reviewer for this comment as our approach is indeed complementary to the work in [19]. Where [19] is restricted to single molecules, as it is the single energy level transition that leads to the measured anti-bunching, our approach can measure the fluorescent decay of an ensemble of fluorophores. Furthermore, as we are not restricted to a single transition, our FL-HOM approach is able to resolve more complicated decay profiles that result in e.g. bi-exponential lifetimes as discussed by one of the other reviewers although this is not demonstrated in

the current work. To address this in the manuscript, we have added the following to the discussion section:

“We also note that as our method directly measures the full temporal distribution of the fluorescent light it allows also for the measurement of bi-exponential and more generalised fluorescent decays. It is also not restricted to measurements of single molecules and in this way, is complimentary to techniques such as those shown in [19].”

as well as correcting the description of [19] in the introduction:

“Fluorescence lifetime can also be measured by monitoring the photon anti-bunching when pumped by two femtosecond pulses with a relative temporal delay [19], this approach is however restricted to isolated single molecules. Photon bunching has also been used for higher densities of molecules [20, 21] with schemes that produce an auto-correlation of the fluorescence signal.”

We note that in principle there is no fundamental reason why fluorescence from a single molecule cannot be measured with the FL-HOM technique, as the reviewer suggests it is simply matter of photon budget although this may not be as restricting as one might first imagine. As the reviewer mentions, in our case we require quite stringent spectral filtering to ensure indistinguishability between the measured photons and thus a high interference visibility but this need not be the case in general. A broadband source could be used for the reference pulse e.g. using a supercontinuum laser, and spectrally filtered to match the emission spectrum of the fluorophore. In this case the requirement of the spectral filtering of the fluorescence is removed. For optimal visibility the reference pulse duration should approximately match the lifetime (see Fig. 4 of the manuscript), however one could choose a supercontinuum source that matches the lifetime of the molecule of interest. We also note that a single molecule will emit into a single spatial mode, removing the need for the mode selection we perform with single mode fibers. So although the photon numbers will be lower than an ensemble of molecules in solution (which is always true for single molecules) there is no reason why all of the fluorescent photons can't be used in the FL-HOM approach.

The reviewer raises an interesting question about the fluorophore concentrations our approach is relevant for. Although we discuss above how a single molecule FL-HOM measurement would be possible, it would require a new setup to explore designed specifically for this task and for the properties of the reference photon to be carefully chosen. We would like to add that it is the freedom to choose an arbitrary reference photon that is at the heart of this technique and is what allows our high temporal resolution. Given these points, we believe this should be the focus of a separate future study.

Besides these points I have to say that I like this approach a lot and it seems to me like an innovative idea. However, the authors might also point out the necessary count rate to perform this measurement and as mentioned above it seems to me that this approach is not applicable to single molecules, due to the limited photon budget available. But maybe the authors can prove me wrong on this, which would strengthen the applicability.

In summary I like this idea and I hope that the authors have very good arguments to overcome this problem or I have to suggest publication in a more specialized journal for spectroscopy.

We would like to take the opportunity to thank the reviewer for their positive comments and we hope we have addressed their concerns satisfactorily.

II. REVIEWER 2

The authors propose and demonstrate a novel fluorescence lifetime measurement technique based on photon-bunching statistics with Hon-Ou-Mandel (HOM) interferometer. Sub-picosecond time resolution is achieved and an exciting application of contactless nanorheology in low-viscosity fluids is provided. Overall, I find the work very exciting and highly relevant, and it should be considered for publication in Nature Communications but only after the authors carefully address the following concerns and questions.

1. The Introduction is rather weak, and it does not make a good case for the need of picosecond fluorescence lifetime measurement. The authors should expand their Introduction far beyond just saying “However, lifetimes can shorten down to 10’s of femtosecond depending on the nature of the fluorophore [1].”

We agree that the introduction could be stronger in this regard. Within the discussion section there is specific reference to new markers that could be more widely adopted using this technique as well as mention of transitions that couple to vibrational modes with sub-picosecond lifetimes that could be investigated. We also discuss how plasmonic

devices can increase the fluorescence yield whilst truncating the lifetime to sub-picosecond timescales. We have now adapted the introduction to mention these factors.

“However, lifetimes can shorten down to 10’s of femtoseconds depending on the nature of the fluorophore [1]. As such, access to higher resolutions enables the use of femtosecond scale fluorophores [2,3], coupling to plasmonic devices which can increase quantum yield [4,5], and exploration of vibrational and rotational modes that act on the femtosecond scale [6]. ”

2. Comparison between their FL-HOM and existing sub-picosecond fluorescence lifetime measurement techniques is insufficient.

a. For its comparison with nonlinear optical gating, the authors claim that “FL-HOM does not rely on any optical nonlinear components and is well suited for low photon numbers.” While the reviewer agrees that their FL-HOM does not require a high-power gate (or reference) pulse, it is not clear whether FL-HOM is indeed more suited in photon starved conditions. As the authors mentioned in the Introduction, up to 50% nonlinear conversion efficiency can be achieved given a high-power gate pulse. The main issue of nonlinear optical gating is the low photon efficiency due to the large mismatch between the gate pulse duration and the fluorescence lifetime. From this perspective, it seems like FL-HOM has a similar issue as the authors showed in Figure 4: there is a trade-off between time resolution and interferometric visibility (and thus measurement sensitivity). The authors should provide more justifications including some photon budget calculation to support their argument.

We apologise for the miscommunication with regards to this point. Here, we wished to state that the FL-HOM approach is advantageous compared to optical nonlinearities in that it does not require high laser powers (for either the fluorescence or reference pulse), does not require a nonlinear crystal, and is not restricted by any phase-matching constraints. We note that maximum power used for the excitation pulse was at most 10 mW whilst the reference pulse was always attenuated to ~ 200 photons per second. We also wished to state that it is able to operate at low photon numbers but make no claims that the efficiency is any higher. We have amended this statement towards the end of the introduction as follows:

“Our approach, which we call Fluorescence Lifetime Hong-Ou-Mandel (FL-HOM), offers the benefit of measuring directly the fluorescence lifetime signal by using a reference photon or laser pulse that is much shorter than the fluorescence lifetime. FL-HOM is able to operate at low photon numbers not only in the fluorescence pulse, but also in the reference pulse, and does not rely on any optical nonlinear components.”

b. Similarly, it is not acceptable that the authors only have a sentence that compares their FL-HOM with FL-HBT [19-21]. First of all, it is the reviewer’s understanding that their claim “However, this approach provides the auto-correlation of the fluorescence signal, which requires the solution of an ill-posed inverse problem to recover the actual lifetime.” only applies to [20, 21] but not to [19]. In fact, it is stated in [19] that fluorescence lifetime can be measured without the need for any deconvolution. Naively, the reviewer would think the authors’ FL-HOM has an edge over FL-HBT [19] because their FL-HOM can be applied to fluorophores with complicated energy-level diagrams. If this is true, the next question is whether FL-HOM can be applied to study double exponential fluorescence lifetime. The reviewer does not expect new measurements, but some discussions will come a long way.

We thank the reviewer for this comment as we did not mean to imply that the approach taken in [19] requires deconvolution, as indeed stated in the paper. Instead, as the reviewer alludes to, the approach in [19] is restricted to a single energy level transition, otherwise antibunching would not be observed. Our approach on the other hand, not only extends to many fluorescent molecules, but also to multiple transitions of different lifetimes leading to double exponential decays as the reviewer states. To reflect this, we have altered the appropriate section of the introduction: “Fluorescence lifetime can also be measured by monitoring the photon anti-bunching when pumped by two femtosecond pulses with a relative temporal delay [19], this approach is however restricted to isolated single molecules. Photon bunching has also been used for higher densities of molecules [20, 21] with schemes that produce an auto-correlation of the fluorescence signal. Recovering the actual lifetime from the auto-correlation requires the solution of an ill-posed inverse problem [22].”

We have also added the following sentence to the discussion:

“We also note that as our method directly measures the full temporal distribution of the fluorescent light it allows also for the measurement of bi-exponential and more generalised fluorescent decays. It is also not restricted to measurements of single molecules and in this way, is complimentary to techniques such as those shown in [19].”

3. The authors show three FL measurements of different fluorophores in Figure 3 and temperature-dependent FL of 4-DASPI in Figure S1. How do their FL-HOM measurement results compare to those using existing methods? Without references, it is hard to evaluate the accuracy of their new technique.

We do appreciate the importance of comparing the lifetimes we retrieve to those obtained with other techniques as it is indeed difficult to judge whether the FL-HOM approach provides reliable results. We have searched extensively in the literature and found only a few examples which could be considered a fair comparison. Reference [23] of our manuscript shows a measurement of 4-DASPI in water, quoting a lifetime of 11 ps where we find 7.22 ps. We attribute this small difference to potential differences in pH, temperature and emission wavelength which are not quoted in the study and therefore could produce a different lifetime as well noting that the IRF in this measurement was substantially longer than the lifetime. We would like to emphasise that measurements of lifetimes on this scale are challenging with pre-existing methods and this is why examples are scarce in the literature. This is exactly what forms the motivation for our work.

There are a number of other studies which examine 4-DASPI in ethanol which is known to produce a longer lifetime (see discussion with Reviewer 3, point 2). We conducted an equivalent measurement finding a lifetime of 65.2 ± 1.2 ps. This is in close agreement with [Sibbett & Taylor, *Opt Comms*, Vol 46, 1, 1983, pg 32-36] quoting a value of 62 ps.

Following the above discussion, we have added this information to the manuscript and the figure below to the supplementary.

“Figure 2(a) also shows a fit to our model given by Eq.(3) and the measured IRF, from which we recover a lifetime of 7.22 ± 0.04 ps. Previous work in [23] measured a lifetime of 11 ps for 4-DASPI in water. Whilst this is relatively close to our measurement, we attribute our shorter lifetime to potential differences in the temperature, pH, and emission wavelength chosen, all of which can affect the lifetime. To help validate our approach, we also conducted a measurement of 4-DASPI in ethanol which is known to produce a longer lifetime. From this we observed a lifetime of 65.2 ± 1.2 ps which is in close agreement with previous measurements [24] (see Supplementary Material).”

4. According to the derivation, Eq (3) is valid only for the case $\sigma \ll \mu$. On the other hand, the optimum visibility happens when $\sigma = 0.7 * \mu$ as shown in Figure 4b. Now, one can argue that σ should not be too much smaller than μ for a reasonable measurement signal-to-noise ratio. For this reason, the reviewer thinks it is important to at least include the first-order perturbation to their analytic model to understand the effect of finite σ/μ .

It is true that the Eq.(3) is only valid for the case when $\sigma \ll \mu$ as also stated in the main text after the equation. We wish to emphasise that we use this approximation only to illustrate how the lifetime can be estimated directly from the measured number of coincident events, C , in the case where the IRF is short. This can be done by tail fitting, for example, as shown in Fig. 3. Where the IRF/lifetime is not within this limit other approaches must be relied on e.g. using numerical deconvolution. This is well known for standard FLIM measurements, however our FL-HOM approach pushes this limit to be valid for shorter lifetimes by utilising a shorter effective IRF. We wish to also clarify that Fig. 4b was calculated numerically without any approximation relying on $\sigma \ll \mu$ i.e. using Eq.(S10). Of the data we show in manuscript, only the tail fitting in Fig. 3 relies on this limit.

We believe it to be crucial to state when this limit is valid in the specific case of our measurements. For example, in Fig. 3 it can qualitatively be discerned that there is a clear linear slope for the 4-DASPI sample, indicating that this approximation is most likely valid, however this is not necessarily true for the Allura Red sample. We have therefore validated the lifetimes we found through the tail fit of our Allura Red sample against our MCMC based

lifetime retrieval that incorporates the distinct shape of the IRF resulting in a 1.5 ps lifetime. Subsequently we have added the following to the manuscript.

“From the analysis of these measurements we retrieve lifetimes of 6.23 (4-DASPI), 1.60 (Allura Red) and 3.66 (Pincyanol Iodide) picoseconds. **As we note that the lifetimes measured here are close to our effective IRF (see below) we validated our shortest lifetime measurement using the Allura Red against our MCMC retrieval. From this, we found a lifetime of 1.51 ps indicating that simple tail fits provide a reasonable lifetime estimation at these short timescales.**”

We have also added the duration of our measured IRFs for the two spectral filters which we realised were missing from the document.

Figure.4(a) shows the normalised number of coincident events for the Allura Red sample with both a 0.6 nm and a 10 nm bandpass filter, keeping all other conditions fixed. **We measure the IRFs with these two filters to be 2.08 ps and 0.17 ps FWHM respectively (See Methods).**

We agree with the reviewer that including a first order perturbation would help describe the scenario more completely, however the theory is formulated in a way where perturbation theory is not readily accessible due to the well known non-trivial perturbative expansion of the error functions, as well as some of the other terms, in Eq.(S11). Nonetheless, it is clear from the numerical evaluation that the coincidence dip will take on more of a Gaussian character as σ approaches μ , as would be expected physically.

5. Another question about Figure 4b. It is the reviewer’s understanding that the maximum visibility does not reach 0.5 because of the pulse shape mismatch between the reference (Gaussian) and the fluorescence signal (exponential). If this is true, have the authors considered other reference pulse shape to get closer to 0.5 visibility?

This is an excellent question and one that we have discussed amongst ourselves on numerous occasions. Indeed, the biggest factor affecting our current visibility is the discrepancy between the pulse shapes. An intuitive solution to this might be to use a pulse shaper, made of dispersive elements and a spatial light modulator, to match the reference pulse to the expected fluorescence. However, we do not believe that interference between a coherent chirped pulse and the incoherent fluorescence would take the same form. We plan on investigating this possibility further but we believe the study to be too extensive to include in the current manuscript.

6. For Figure 5, there are only three data points in the low viscosity regime (1-2 mPa \times s) where the FL-HOM is most powerful at. The authors should take a few more data points to strengthen their error analysis for this important low viscosity regime.

Whilst we do only have 3 data points within this regime we would like to emphasise that the full range of viscosities we explore (up to 5 mPa \times s) are at the lower limit of viscosities for cellular environments. The viscosity in cells can extend up to 400 mPa \times s with anything above 50 mPa \times s being considered very high. Our technique is therefore probing the lower 1-5% of the full dynamic range of viscosities one would expect to see in a cell. We would like to apologise for the confusion with regards to this point and have added the following statement to the manuscript with references supporting these numbers.

“We used 4-DASPI dye in water and glycerol/water mixtures to explore a range of low viscosity values (i.e., of the order of mPa \cdot s), whilst maintaining a constant pH and temperature (see Methods). **The viscosity within cellular environments can lie between 1 - 400 mPa \cdot s [33-35], here we focus on the lower 1-5% of this range to demonstrate the high sensitivity of our technique.**”

We have also added a new measurement point to Fig.5 that sits at the lower end of this scale, shown in red below.

7. At the end of the paper, the authors claim that a PDC source can better match the fluorescence emission wavelength and provide a factor of 2 increase in the HOM visibility. Didn’t the authors already match the wavelength by using narrowband filters? It is not clear why PDC source will help and where the factor of 2 comes from.

We thank the reviewer for this comment as we appreciate the way this sentence has been phrased is a little misleading. There are two components at play here. Firstly, an SPDC source would provide greater flexibility in fluorescence wavelength as the downconverted pairs could be tuned away from wavelength degeneracy by changing the phase matching conditions e.g. by tilting the crystal used for generation. This statement was intended to discuss tunability of our approach rather than the visibility. Secondly, and separate from the first point, a single photon source would provide a factor 2 \times increase in the visibility due to the different photon arrival statistics, not the spectral properties. Previous studies have shown that interference between a single photon source and e.g. a weak coherent state can reach a visibility of 1. See, for example, Rarity et al. 2005 J. Opt. B: Quantum Semiclass. Opt.

7 S171 <https://iopscience.iop.org/article/10.1088/1464-4266/7/7/007>. We apologise for missing this reference as we believe it helps answer the reviewer's query. We have amended this paragraph in the manuscript.

“Finally, our experiments are implemented with a classical reference laser pulse. One could use a single photon generated for example by a parametric downconversion (PDC) or a quantum dot source. It has previously been shown that the visibility of the HOM interference between a single photon source and, for example, an attenuated laser [17] can approach unity where an equivalent measurement with classical light would be limited to a $g^{(2)}$ reduction of 0.5. Furthermore, a PDC source provides greater flexibility in terms of tuning the reference wavelength to the fluorescence emission by operating at non-degenerate photon pair wavelengths controlled through the phase-matching of the generation crystal.”

III. REVIEWER 3

The authors report on a novel method to measure fluorescence decays with exquisite picosecond time resolution. It is based on the Hong-Ou-Mandel effect which causes two photons incident onto a beamsplitter at the same time from different directions to be both reflected and transmitted in the same direction, i.e. together, as illustrated here

Michał Jachura and Radosław Chrapkiewicz, “Shot-by-shot imaging of Hong–Ou–Mandel interference with an intensified sCMOS camera,” *Opt. Lett.* 40, 1540-1543 (2015) <https://doi.org/10.1364/OL.40.001540>

In the present manuscript, one photon comes from the fluorescence decay, and one photon comes from the excitation pulse. One of them is registered in one detector, whereas the other detector is guaranteed not to detect a photon. It may be worth a mention that only one of the two Hong-Ou-Mandel photons is detected, to make this clearer.

Anyway, this causes a dip in the second order correlation function g_2 , if I understand this correctly. The delay between excitation pulse photon and fluorescence photon is changed, and the depth of the dip in g_2 is plotted versus the delay, and this represents the fluorescence decay. As the optical width of the excitation pulse is femtoseconds, a picosecond time resolution can be achieved with this method.

This is one of the highest time resolutions reported, similar to streak cameras

ANTHONY J. CAMPILLO AND STANLEY L. SHAPIRO, Picosecond Streak Camera Fluorometry-A Review *IEEE JOURNAL OF QUANTUM ELECTRONICS*, VOL. QE-19, NO. 4, APRIL 1983 585, doi: 10.1109/JQE.1983.1071909

and a recently reported short linear super-conducting nanowire

Korzh, B., Zhao, QY., Allmaras, J.P. et al. Demonstration of sub-3 ps temporal resolution with a superconducting nanowire single-photon detector. *Nat. Photonics* 14, 250–255 (2020). <https://doi.org/10.1038/s41566-020-0589-x>

Hadfield, R.H. Superfast photon counting. *Nat. Photonics* 14, 201–202 (2020). <https://doi.org/10.1038/s41566-020-0614-0>

The authors put this approach to use by studying short fluorescence lifetime fluorophores, including a fluorescent molecular rotor. They show its fluorescence lifetime increase in the 10s of picosecond range with increasing viscosity, as expected for these types of fluorophores.

This is an ingenious idea reminiscent of pump probe approaches which have been proposed for fluorescence lifetime measurements

Buist, A.H., Müller, M., Gijsbers, E.J., Brakenhoff, G.J., Sosnowski, T.S., Norris, T.B. and Squier, J. (1997), Double-pulse fluorescence lifetime measurements. *Journal of Microscopy*, 186: 212-220. <https://doi.org/10.1046/j.1365-2818.1997.2090773.x>

and FLIM

MÜLLER, M., GHAUHARALI, R., VISSCHER, K. and BRAKENHOFF, G. (1995), Double-pulse fluorescence lifetime imaging in confocal microscopy. *Journal of Microscopy*, 177: 171-179. <https://doi.org/10.1111/j.1365-2818.1995.tb03547.x>

The manuscript is well written and straight-forward to follow, with good contextualisation and discussion of alternative methods to measure fluorescence decays.

This is an excellent piece of work that will be of interest to fluorescence spectroscopists and probably microscopists, the FLIM community and the applied quantum optics research community.

We thank the reviewer for their positive comments and their note on the wide scope of interest we hope our work will gain.

I only have some minor comments:

1) Is the Nikon 60x 0.7 NA an air objective?

Yes, an air objective was used. We have added this detail into the paper.

“This is frequency-doubled in a BBO crystal to 520 nm and focused on to the fluorescent sample using a Nikon 60×, 0.7 NA microscope **air** objective.”

2) Prior DASPI and pinacyanol decay work can be found here

C.J. TREDWELL and CM. KEARY, PICOSECOND TIME RESOLVED FLUORESCENCE LIFETIMES OF THE POLYMETHINE AND RELATED DYES, *Chemical Physics*, Volume 43, Issue 3, 1 November 1979, Pages 307-316 [https://doi.org/10.1016/0301-0104\(79\)85199-X](https://doi.org/10.1016/0301-0104(79)85199-X)

W. Sibbett, J.R. Taylor, Passive mode locking in the blue spectral region, *Optics Communications* 46(1), 32-36, 1983, [https://doi.org/10.1016/0030-4018\(83\)90025-1](https://doi.org/10.1016/0030-4018(83)90025-1)

Michael Maus, Els Rousseau, Mircea Cotlet, Gerd Schweitzer, Johan Hofkens, Mark Van der Auweraer, and Frans C. De Schryver, Arnd Krueger, New picosecond laser system for easy tunability over the whole ultraviolet / visible / near infrared wavelength range based on flexible harmonic generation and optical parametric oscillation *Rev Sci Instrum* 72, 36-40 (2001) <https://doi.org/10.1063/1.1326930>

Do the authors' lifetime values agree with the ones quoted in these references?

The reviewer raises a valid question and indeed we have looked in the literature for previous studies we can compare our measurements to. In the three cases the reviewer lists, a methanol/ethanol solution was used for the DASPI dye whereas our dye was in water (and later a water-glycerol mix). These alcohol solutions are known to produce a longer lifetime, a comparative measurement can be found here https://chromacitylasers.com/wp-content/uploads/2021/04/Picosecond_Lifetimes_App_Note_v1_Jun_2020.pdf. We do note, however, that both this measurement and that found in [23] of our manuscript found a lifetime of order 11-12 ps for DASPI in water whereas we measure approximately 6-7 ps. We attribute this to potential differences in environmental parameters (e.g. temperature, pH), and emission wavelength and thus don't necessarily translate to our work. We also wish to highlight

that both of these previous measurements were performed with IRFs substantially longer than the lifetime (34/50 ps) and therefore in a regime where it is notoriously difficult to recover an accurate lifetime and that this is precisely the motivation for our work.

We have also conducted a measurement of our 4-DASPI dye in ethanol from which we obtained a lifetime of 65 ps. This is in close agreement with the measurement of Sibbett and Taylor who observed a lifetime of 62 ps. We have added the following to the manuscript to reflect this comparison with literature as well as adding the DASPI in ethanol measurement to the supplementary.

“Figure 2(a) also shows a fit to our model given by Eq.(3) and the measured IRF, from which we recover a lifetime of 7.22 ± 0.04 ps. Previous work in [23] measured a lifetime of 11 ps for 4-DASPI in water. Whilst this is relatively close to our measurement, we attribute our shorter lifetime to potential differences in the temperature, pH, and emission wavelength chosen, all of which can affect the lifetime. To help validate our approach, we also conducted a measurement of 4-DASPI in ethanol which is known to produce a longer lifetime. From this we observed a lifetime of 65.2 ± 1.2 ps which is in close agreement with previous measurements [24] (see Supplementary Material).”

In the case of Pinacyanol, Tredwell and Cleary list a lifetime value of Pinacyanol Chloride in ethanol of 13 ps. Our measurement of Pinacyanol Iodide in methanol retrieved a lifetime of 3.66 ps. We believe the differences in both the dye and the solvent do not guarantee this comparison to be reliable.

3) p3, first column, line 3 – the resolution is the “time resolution”, I presume? Maybe explicitly say “time resolution”

Yes, the reviewer is correct. We have changed this to “achievable temporal resolution.”

4) p3, second column, line 4, full time distribution, fig 2c. How many measurements is this? The integral of the distribution? A few hundred?

Yes, the total number of measurements is the integral of the distribution. 260 measurements were used, this is stated in the figure caption but we have added it also to the main text.

“A full distribution of lifetimes derived under these conditions (i.e., 3.5 s total acquisition time, 260 individual measurements) is shown in Fig.2(c)...”

5) p3, second column, line 14, full time distribution, fig 2d. In TCSPC, the standard deviation depends on the square root of the number of counts (or the time) due to Poisson statistics. So doubling the acquisition time in TCSPC only improves the standard deviation by a factor of $\sqrt{2}$, whereas here it halves it. Is this right? If so, this would be worth a mention.

The reviewer’s comment has prompted us to re-evaluate our data analysis as we also found this surprising. During this, we noticed a mistake related to how we sample our data from the numerous individual measurements we obtained, discussed in Section IV of the SM. We have since rectified this and now find an improvement that scales as the square root of the measurement time as expected. We note that this is not only true for Poissonian statistics, but in fact any independent and identically distributed variable. During this process, we also found that the uncertainty of our lifetime estimations improved. We have updated the relevant numerical values in the manuscript and updated Fig.2

(shown below).

We would like to thank the reviewer for this comment as it also led to an interesting internal discussion amongst the authors about the expected statistical uncertainty of the number of measured coincident events if the photon number at each of the two detectors is limited by Poisson noise. We have included our findings in a new section of the supplementary document showing that one would expect to observe a $\frac{N}{\sqrt{1+2N}}$ scaling where N is the average photon number.

To reflect these additions/changes, we have altered the text in the main document as follows:

“In Fig.2(d) the red line indicates a gradient of -0.5 i.e. a line proportional to the square root of the total measurement time which we expect to observe given that we have a set of independent and statistically equivalent measurements. It is also useful to determine how the number of coincident events themselves scale as a function of the measurement time. We evaluate this (see SM) finding that the Signal to Noise Ratio improves proportionally to $N/\sqrt{1+2N}$. For $N \gg 1$ the SNR $\sim \sqrt{N}/\sqrt{2}$, indicating that coincidence counting brings a penalty of $\sqrt{2}$ in the SNR compared to direct intensity measurements.”

6) p4, second column, short DASPI lifetime. As $\phi = k_r \tau$, with ϕ the fluorescence quantum yield, k_r the radiative rate constant and τ the lifetime, short lifetime fluorophores usually have a low fluorescence quantum yield, i.e. they are dim. Long lifetime fluorescent molecular rotors like bodipys are not only easier to measure, but they are also brighter.

It is indeed true that, as a general rule, shorter lived fluorophores have a lower quantum yield. Within the scope of our work, we wished to illustrate the high temporal resolution of the FL-HOM approach which is most clear using short lifetime probes. We emphasise that this approach is complementary to TCSPC based FLIM which is more suitable for dyes such as BODIPYs that have lifetimes on the nanosecond scale. Nevertheless, this is an important point that we feel should be reflected in the text. We have a statement to this effect in the discussion section.

“Although longer lifetime fluorophores generally have a greater quantum yield than shorter lifetime probes, interaction with plasmonic modes have also shown a strong increase in the fluorescence intensity at the expense of shortening the lifetime [37,38].”

7) p5, fig 5. Calibration plots for fluorescent molecular rotors are often presented on a double logarithmic scale (as log lifetime vs log viscosity) with a straight-line fit yielding a gradient around 2/3. Such a plot would also be useful here, maybe as an inset.

We agree that this is potentially useful to readers and have added this to the Supplementary Materials. We also

include the fit the reviewer suggested although we retrieve a gradient of 1.38. We note that the expected gradient of 2/3 from the Förster-Hofmann relation [Förster, T. and Hoffmann, G. (1971) *Die Viskositätsabhängigkeit der Fluoreszenzquantenausbeuten einiger Farbstoffsysteme. Zeitschrift für Physikalische Chemie, Vol. 75 (Issue 1 2), pp. 63-76.*] is known to only be valid for intermediate viscosities. In the low viscosity regime explored in our work this no longer holds [Kuimova (2012) *Phys. Chem. Chem. Phys.*, 14, 12671-12686].

8) p5, column 2, line 20 “lifetime decay of fluorescence” doesn’t make any sense, the authors probably mean “fluorescence decay”, so that it should read something like “. . . can be resolved within the duration of the fluorescence decay. . .”

Yes we agree and thank the reviewer for this comment. The sentence has been changed as the reviewer suggested.

9) p5, column 2, Methods, fluorophore concentration. mM concentrations are very high, the solutions must have been noticeably coloured. According to

Chandrasekhar, S. 1943. Stochastic problems in physics and astronomy. *Rev. Mod. Phys.* 15:1–89. <https://doi.org/10.1103/RevModPhys.15.1>

the average distance between individual fluorophores is $0.55/[c]^{1/3}$ with $[c]$ the concentration in particles/volume. At 5 mmol/l, this yields 3.8 nm. The average distance the fluorophore diffuses while in the excited state is given by $l = \sqrt{2 D \tau}$, where D is the translational diffusion coefficient, given by $D = kT/6 \pi \eta r$, with k Boltzmann constant, T absolute temperature, η viscosity and r the radius of the fluorophore. For $T=293$ K, $\eta = 1$ cp, $r = 0.5$ nm, we have $D = 430$ nm²/μs, in good agreement with experimental values for Coumarin 343 (550 nm²/μs) and rhodamine 6 (400 nm²/μs) according to Viplove Tyagi et al 2022 *Methods Appl. Fluoresc.* 10 044007, <https://iopscience.iop.org/article/10.1088/2050-6120/ac896c>

Thus $l = 0.1$ nm, and, on average, the fluorophores do not interact during the lifetime of the excited state, and re-absorption of the emitted fluorescence is probably also low due to the large Stokes shift of DASPI. Fig 5 should then really only represent viscosity sensitivity, and not interaction or self-absorption. This may perhaps be worth a brief discussion?

Indeed, the samples used were noticeably coloured and as such we believe it to be important to rule out any other potential factors that could be at play. To that end we have added the following section to the Supplementary Material to discuss the probability of the fluorophores interacting whilst being in the excited state.

“We note that the concentration of fluorescent dyes used in this study (or order mM) are relatively high and it is thus important to discern whether or not interaction between fluorescent molecules occur which can potentially affect the lifetime. Firstly, we estimate the average distance between fluorescent molecules as

$$x = \frac{0.55}{n^{-1/3}} \quad (1)$$

following [Chandrasekhar] where n is the concentration in particles per unit volume. For our highest concentration sample at ~ 5 mM, this yields $x \approx 3.8$ nm. Next, we evaluate the average distance a fluorescent molecule moves whilst in the excited state, which we define using the lifetime. Assuming Brownian motion, this can be calculated as

$$\delta x = \sqrt{2D\mu} \quad (2)$$

with D as the translational diffusion coefficient.

$$D = \frac{k_B T}{6\pi\eta r} \quad (3)$$

where k_B is the Boltzmann constant, T the absolute temperature, η the viscosity, and r the average particle radius [Chandrasekhar]. Taking $T = 293$ K, $\eta = 1$ mPa·s (corresponding to water), and $r = 0.5$ nm, we find that $D = 430$ nm²/μs. This results in a δx of 0.1 nm. As this is significantly smaller than the average distance between particles, it is reasonable to conclude that there is no interaction between excited fluorophores and neighbouring molecules.”

Furthermore, we have added a statement in the Methods section to reflect the reviewer’s comment on the Stokes shift which we agree limits re-absorption, as well as reference the new section in the Supplementary.

“Fluorescent dye sample preparation. 4-DASPI (Sigma-Aldrich 336408) and Allura Red AC (Sigma-Aldrich 458848) were dissolved in purified water at concentrations of ≈ 2.7 3mM and ≈ 5.04 mM respectively, and the Pina-cyanol iodide (Alfa Aesar H31540) was dissolved in MeOH (Sigma-Aldrich 1.06002) at the concentration of ≈ 0.12 mM. The samples were measured at room temperature, in 10 mm path length UV fused quartz cuvettes (Thorlabs CV10Q35E). *We note that in all cases, the probability of re-absorption in these samples is low due to the relatively large Stokes shift. We also confirm that there is a low probability of interaction between fluorophores whilst in the excited state. A discussion of this can be found in the SM.*”

10) ref 5, details?

Thank you to the reviewer for spotting this, we have revised the citation.

11) SI, line 6 after eq S1b. Do the authors mean eq 1 in the main text?

Yes, this was indeed a typo. Now corrected.

12) SI eq S2. What is R and T?

These are the reflectance and transmittance of the final beamsplitter in the interferometer. We have added this to the SM directly after the equation.

13) Si fig S1, State solvent used to dissolve 4-DASPI

The solvent used was water, as in our DASPI measurements in the main paper. We have added this to the figure caption.

REVIEWERS' COMMENTS

Reviewer #1 (Remarks to the Author):

The authors have addressed my concerns and the concerns of the other reviewers very carefully and satisfactorily. Therefore, I believe that this manuscript is now ready for publication and again I congratulate the authors for their work.

Reviewer #2 (Remarks to the Author):

In my opinion, the authors have carefully and satisfactorily addressed all the reviewers' comments. I would recommend it for publication in Nature Communications.

Reviewer #3 (Remarks to the Author):

The time and effort the authors have spent on revising the manuscript has made it clearer and contextualised it better. The more extensive discussion and the expanded SI is also helpful for a readership from a wide range of backgrounds. It is also nice to read that some of the reviewers' comments have prompted discussion among the authors. All in all, this is how peer-review is supposed to work, isn't it!?

In the SI, in the line before eq S20, it should read "Brownian" (not Browninan).

I look forward to seeing the work published.